# Synergism between IL-33 and MRGPRX2/FcεRI Is Primarily Due to the Complementation of Signaling Modules, and Only Modestly Supplemented by Prolonged Activation of Selected Kinases

**DOI:** 10.3390/cells12232700

**Published:** 2023-11-24

**Authors:** Kristin Franke, Zhuoran Li, Gürkan Bal, Torsten Zuberbier, Magda Babina

**Affiliations:** 1Fraunhofer Institute for Translational Medicine and Pharmacology ITMP, Immunology and Allergology IA, 12203 Berlin, Germany; kristin.franke@charite.de (K.F.); zhuoran.li@charite.de (Z.L.); guerkan.bal@charite.de (G.B.); torsten.zuberbier@charite.de (T.Z.); 2Institute of Allergology, Charité—Universitätsmedizin Berlin, Corporate Member of Freie Universität Berlin and Humboldt Universität zu Berlin, Hindenburgdamm 30, 12203 Berlin, Germany

**Keywords:** mast cells, IL-33, FcεRI, MRGPRX2, TNF-α, p38, ERK, JNK, PI3K, NF-κB

## Abstract

Skin mast cells (MCs) express high levels of MRGPRX2, FcεRI, and ST2, and vigorously respond to their ligands when triggered individually. IL-33/ST2 also potently synergizes with other receptors, but the molecular underpinnings are poorly understood. Human skin-derived MCs were stimulated via different receptors individually or jointly in the presence/absence of selective inhibitors. TNF was quantified by ELISA. Signaling cascades were studied by immunoblot. TNF was stimulated by FcεRI ≈ ST2 > MRGPRX2. Surprisingly, neither FcεRI nor MRGPRX2 stimulation elicited NF-κB activation (IκB degradation, p65 phosphorylation) in stark contrast to IL-33. Accordingly, TNF production did not depend on NF-κB in FcεRI- or MRGPRX2-stimulated MCs, but did well so downstream of ST2. Conversely, ERK1/2 and PI3K were the crucial modules upon FcεRI/MRGPRX2 stimulation, while p38 was key to the IL-33-elicited route. The different signaling prerequisites were mirrored by their activation patterns with potent pERK/pAKT after FcεRI/MRGPRX2, but preferential induction of pp38/NF-κB downstream of ST2. FcεRI/MRGPRX2 strongly synergized with IL-33, and some synergy was still observed upon inhibition of each module (ERK1/2, JNK, p38, PI3K, NF-κB). IL-33’s contribution to synergism was owed to p38 > JNK > NF-κB, while the partner receptor contributed through ERK > PI3K ≈ JNK. Concurrent IL-33 led to slightly prolonged pERK (downstream of MRGPRX2) or pAKT (activated by FcεRI), while the IL-33-elicited modules (pp38/NF-κB) remained unaffected by co-stimulation of FcεRI/MRGPRX2. Collectively, the strong synergistic activity of IL-33 primarily results from the complementation of highly distinct modules following co-activation of the partner receptor rather than by altered signal strength of the same modules.

## 1. Introduction

IL-33 is a potent alarmin, that, in the cutaneous environment, is released by several cells, including keratinocytes, endothelial cells, and fibroblasts following infection, microbial products, or trauma and can contribute to inflammatory dermatoses (e.g., atopic dermatitis, psoriasis, urticaria) as well as other inflammatory conditions [1,2,3,4,5,6,7,8]. Direct intradermal application of IL-33 gives rise to leukocyte recruitment and skin inflammation [9]. IL-33 signals via the ST2/IL1RacP complex, a receptor abundantly expressed by MCs even in the steady state [4,10,11,12]. As a consequence, MCs are among the most important targets of IL-33 and are considered crucial amplifiers in IL-33-induced inflammation [13,14,15,16,17].

IL-33 effects on MCs are manifold. For instance, the alarmin supports MC survival and proliferation in mouse and man, suggesting assistance in long-term maintenance of the MC compartment [18,19,20,21]. It also increases MC histamine contents through augmented expression of histidine decarboxylase [20,22]. Moreover, IL-33 primes MC degranulation (without degranulating MCs on its own) and activates cytokines by itself and in combination with other stimuli [16,21,23,24,25,26,27,28,29,30].

Conversely, IL-33 can also interfere with MC stimulation by downregulating receptor expression and modulation of the signaling apparatus in a chronic scenario [20,28,31,32]. Therefore, MCs possess instruments to protect themselves from IL-33-mediated inflammation on prolonged exposure. Notwithstanding, the cytokine is of chiefly pro-inflammatory nature in acute settings, in which MCs are activated to fight infections and other dangers.

MAS-related G protein-coupled receptor-X2 (MRGPRX2) has leapt into prominence in the MC field as the major receptor of IgE independent MC activation [33,34,35,36,37,38,39,40,41,42,43,44,45]. Across tissues, it is particularly relevant in the skin due to its predominant expression in cutaneous MCs over other MC types [11,46,47,48,49]. The ensuing symptoms from MRGPRX2 activation closely match those elicited by allergic MC stimulation.

We recently reported on potent co-operation between IL-33 and MRGPRX2 in skin MCs [50]. However, the mechanisms behind this remarkable synergy remain to be established.

Indeed, a major hallmark of IL-33 is its potent co-operation with other receptor systems. Synergism has been described for various stimulatory receptors and agents, including KIT, C5aR, nucleotide receptors (e.g., P2X7), bacillus Calmette–Guérin, human rhinovirus, as well as the canonical route involving FcεRI aggregation [26,51,52,53,54,55,56,57]. So far, the FcεRI/IL-33 connection has found the greatest interest. In our recent work, IL-33 potentiated stimulation of skin MCs initiated via this canonical route but also by the MRGPRX2-driven pathway [32,50]. Both degranulation and cytokine responses were affected, though co-operation was more prominent at the level of cytokines [32,50]. Mechanistically, IL-33 elicited three major cascades in skin MCs, namely, p38, c-Jun N-terminal kinase (JNK), and nuclear factor kappa-light-chain-enhancer of activated B cells (NF-κB), that were relevant to different degrees in cytokine activation by IL-33 alone [50]. However, it is unclear how the synergism is molecularly governed and to what degree these components contribute when MCs are exposed to several stimuli simultaneously.

A priori co-operation could stem from complementation of different modules stimulated by the individual pathways, considering that the nature of signaling pathways can differ across receptors (synergism at the level of module diversity). Alternatively, it could occur as the result of a stronger overall activation of the same modules (synergism at the level of signal strength and/or duration).

Here, we interrogate these options and report that signaling pathways activated by IL-33 are distinct from those elicited by MRGPRX2 or FcεRI, while the latter display great overlap. Co-operativity between IL-33 and MRGPRX2/FcεRI is found to chiefly stem from the richness of complementary modules activated simultaneously.

## 2. Materials and Methods

### 2.1. Cells and Treatments

MCs were isolated from human foreskin tissue as described [32,58,59]. To achieve sufficient cell numbers, each mast cell preparation/culture originated from several (2–15) donors, as routinely performed in our lab [20,32,60,61,62,63].

The skin was obtained from circumcisions, with written, informed consent of the patients or legal guardians and approval by the university ethics committee (protocol code EA1/204/10, 9 March 2018). The experiments were conducted according to the Declaration of Helsinki principles. Briefly, the skin was cut into strips and treated with dispase (Boehringer Mannheim, Mannheim, Germany) at 0.5 mg/mL at 4 °C overnight, the epidermis was removed and the dermis finely chopped, and then digested with 1.5 mg/mL collagenase (Worthington, Lakewood, NJ, USA), 0.75 mg/mL hyaluronidase (Sigma, Deisenhofen, Germany), and DNase I at 10 µg/mL (Roche, Basel, Switzerland). The cells were filtered stepwise from the resulting suspension (100 µm and 40 µm strainers). To further purify MCs, anti-human c-Kit microbeads and the Auto-MACS separation device were used (both from Miltenyi Biotec, Bergisch Gladbach, Germany), giving rise to 98–100% MC preparations. The purity of the isolated skin mast cells was verified by acidic toluidine blue staining (0.1% in 0.5 N HCl), as described [64,65].

MCs were cultured in the presence of SCF (100 ng/mL) and IL-4 (20 ng/mL) freshly provided twice weekly when cultures were re-adjusted to 5 × 10^5^/mL. MCs were automatically counted with a CASY TTC (Innovatis CASY Technology, Reutlingen, Germany) [60,61,66].

Experiments were performed 3–4 d after last addition of cytokines and cells were deprived of growth factors (GFs) and FCS (minimal medium) for at least 4 h prior to stimulation for downstream experiments. Each experiment was performed on several individual cultures (given as a “dot” in the dot plots displayed in the figures). For inhibition studies, cells were pre-incubated with SB203580 (p38-Inhibitor; 10 µM), SP600125 (JNK Inhibitor; 10 µM), BAY11-7082 (NF-κB inhibitor; 1 µM), SCH772989 (ERK1/2 inhibitor; 10 µM), and Pictilisib (PI3K inhibitor; 10 µM), all from Enzo Life Sciences, Germany, for 15 min. Inhibitors were used at non-cytotoxic concentrations, as determined previously for skin-derived MCs [67,68] and verified for every new batch. IL-33 was purchased from PeproTech (Hamburg, Germany) and applied at a concentration of 20 ng/mL, as described [32]. To cross-link FcεRIα, AER-37 was used at a concentration of 0.2 µg/mL (anti-FcεRIα-Ab; eBioscience, San Diego, CA, USA). To activate MRGPRX2, c48/80 (Sigma, Steinheim, Germany) or SP (Bachem, Budendorf, Switzerland) were used at 10 µg/mL or 30 µM, respectively, in accordance with previous studies and corresponding to saturating concentrations [59,61,69,70,71,72,73]. Cells were stimulated with the stimuli either singly or in combinations containing IL-33 plus either AER-37 or c48/80 or SP.

### 2.2. ELISA

MCs (at 1 × 10^6^ cells/mL) were treated with inhibitors for 15 min prior to the addition of the stimulus (IL-33, AER-37, c48/80, or SP singly and in combinations versus unstimulated control cells). And 24 h later, supernatants were collected for cytokine measurements. ELISAs were performed according to the manufacturers’ instructions and as described previously [32]. The following kit was used: TNFα (Thermo Fisher Scientific, Berlin, Germany, catalogue number: 88-7346-22).

### 2.3. Immunoblot Analysis

MCs stimulated with the specified agents were collected by centrifugation and immediately solubilized in SDS-PAGE sample buffer and boiled for 10 min. Samples corresponding to equal numbers of cells were subjected to immunoblot analysis. The primary antibodies, all purchased from Cell Signaling Technologies (Frankfurt am Main, Germany), were as follows: anti-pp38 (T180/Y182, #9211), anti-pSAPK/JNK (T183/Y185, #9251), anti-pp65 (S536, #3033), anti-IκBα (#4812), anti-pERK1/2 (T202/Y204, #9251), anti-pAKT (S473, #9271), anti-α-actinin (#6487), and anti-Cyclophilin B (#43603), the latter two serving as loading controls.

A peroxidase-conjugated goat anti-rabbit IgG was used as the detection antibody (Merck, #AP132P). Proteins were visualized by a chemiluminescence assay (Weststar Ultra 2.0, Cyanagen, Bologna, Italy), according to the manufacturer’s instructions and bands were recorded on a chemiluminescence imager (Fusion FX7 Spectra, Vilber Lourmat, Eberhardzell, Germany). Quantification of recorded signals was performed using the ImageJ software (Rasband, W.S., ImageJ, U.S. National Institutes of Health, Bethesda, MD, USA, https://imagej.nih.gov/ij/ (last accessed on 01 September 2023), 1997–2018). Individual intensity values for the detected proteins were normalized to the intensity of the housekeeping proteins cyclophilin B and α-actinin of the same membrane.

### 2.4. Statistics

Statistical analyses were performed using PRISM 9.5 (GraphPad Software, La Jolla, CA, USA). For calculations of differences between more than two groups, the RM one-way ANOVA with Holm-Šídák’s multiple comparisons test was applied (normal distribution) or Friedman test with Dunn’s multiple comparisons test (in case of non-normal distribution). And *p* < 0.05 was considered statistically significant.

## 3. Results

### 3.1. Signaling Modules Differentially Contribute to TNF Generation in Dependence of the Triggering Receptor: Dichotomy between MRGPRX2/FcεRI and IL-33

We selected TNF-α (tumor necrosis factor-α, later abbreviated as TNF) as readout since it is inducible in skin MCs by the three receptor systems in focus, has pathophysiological relevance, and signaling prerequisites underlying its production by single stimuli have been partially clarified [50,58,74,75,76].

In this study, FcεRI-aggregation and ST2 ligation elicited similar levels of TNF, followed by c48/80. As described previously [74,76], SP-stimulated levels were only slightly above control, verifying that distinct MRGPRX2 ligands differ in their potency to activate cytokine expression (Figure 1a and Appendix A). SP, however, elicited cytokine transcripts more potently than the protein [50,60], suggesting that signals to produce and secrete the cytokine protein are inefficiently propagated upon stimulation with SP. The focus of this work is on the released protein product which ultimately makes the function, and especially on the mechanisms behind the synergistic action of distinct receptors.

FcεRI- and MRGPRX2-triggered TNF was nearly abrogated by ERK1/2 inhibition, while IL-33 activated substantial TNF even when ERK activity was suppressed (Figure 1b and Appendix A). Interference with JNK was overall less effective, yet JNKi inhibited FcεRI- and MRGPRX2-elicited TNF, as it also countered the IL-33-driven process even if to a lesser degree (Figure 1c). Significance upon JNKi was reached for the FcεRI-triggered route, but tendencies were visible in the other groups as well (Appendix A). Conversely, FcεRI and MRGPRX2 activation gave rise to robust TNF responses when p38 was rendered nonfunctional, while the response to IL-33 was completely abrogated (Figure 1d and Appendix A). Unexpectedly, the NF-κB pathway did not contribute to FcεRI- or MRGPRX2-stimulated TNF, yet it did so when IL-33 was used as the stimulus (Figure 1e and Appendix A). PI3K made a positive contribution to FcεRI- and MRGPRX2-activated TNF (with lower impact than ERK), while the IL-33 driven process remained unperturbed following PI3K inhibition (Figure 1g and Appendix A). Collectively, the requirements for TNF stimulation match between FcεRI and MRGPRX2, with a rank order of ERK1/2 > PI3K > JNK, and no impact of either NF-κB or p38. The picture is very distinct for IL-33. Here, the strongest effect was found for p38, followed by NF-κB, the rank order thus being p38 > NF-κB > ERK ≈ JNK (whereby only p38 and NF-κB reached statistical significance in the one-stimulus-multiple-inhibitor comparison shown in Appendix A). There was no impact from PI3K. The data suggest the existence of two well-distinguishable routes, one used by allergic/pseudo-allergic stimulation of skin MCs, the other by the alarmin IL-33.

### 3.2. IL-33 Efficiently Activates NF-κB, While MRGPRX2 and FcεRI Barely Activate the NF-κB Module in Skin MCs

The varying contributions of signaling modules to TNF production in dependence of the activated receptor prompted us to explore their stimulation by the three receptors side by side.

As expected [20], IL-33 stimulated a vigorous NF-κB response in skin MCs, including p65 phosphorylation (complete after 2 min and durable) and IκB degradation (minimum reached at ≈15 min, Figure 2, upper-left panel). Neither IgER-crosslinking nor MRGPRX2 were able to elicit a similar response. In fact, IκB degradation was inefficient at best, and no increase in pp65 was noted over baseline (Figure 2, upper-right and lower panels).

We conclude that in human skin MCs, the two degranulation-competent systems, i.e., FcεRI and MRGPRX2, are inefficient activators of the NF-κB pathway. This is a novel and unexpected finding, since NF-κB is typically viewed as a major component of MC stimulation.

### 3.3. ERK and AKT Are More Strongly Activated by FcεRI and MRGPRX2 Vis-à-Vis IL-33, While the Opposite Applies to p38

We previously reported that ERK1/2 and AKT are activated by MRGPRX2 and FcεRI, while among kinases p38 was the most prominent module following IL-33 stimulation [20,76]. Here, we verified the pattern in cells treated with four stimuli side-by-side (Figure 3).

In fact, pERK was prominently and consistently induced following IgER-crosslinking, and MRGPRX2 ligation (Figure 3a) Note that the most prominent signal for MRGPRX2 is often found after 1 min, a time point not included here. ERK activation by IL-33 was only found occasionally, was delayed, and weaker (Figure 3a). Possibly, ERK is not directly activated via IL-33/ST2 alone but requires communication between ST2 and KIT [26], which does not invariably manifest across donors, as previously observed in skin MCs [20].

JNK phosphorylation was only occasionally detectable and modest at best (not shown), as it was also difficult to unequivocally detect pJNK in skin MCs in previous efforts [68,76]. Notwithstanding, this slight phosphorylation of JNK is apparently sufficient to enable functional implication despite its being hardly detectable by immunoblot.

A signal for pp38 was consistently found prior to stimulation (in contrast to pERK and pAKT). Induction above baseline was highly efficient for IL-33, and the signal was persistent (Figure 3a). In contrast, activation above baseline was rather modest downstream of FcεRI and MRGPRX2 (Figure 3a).

The pAKT was virtually not activated by IL-33, but prominently induced via FcεRI and MRGPRX2 (Figure 3b). Again, the maximum for the allergic pathway was at around 15 min, whereas the signal after c48/80 and SP was maximal at 2–5 min (in line with Wang et al. 2022 [76]).

The phosphorylation efficiencies and kinetics confirm our previous findings on MAPKs and AKT but in contrast to previous efforts [50,76], they were inspected simultaneously for all three receptors herein. Together with the insights into NF-κB (dealt with in the previous paragraph), the existence of a dichotomy between IL-33/ST2 on the one hand and FcεRI/MRGPRX2 on the other could be solidified.

### 3.4. Synergism between IL-33 and FcεRI

When stimulated simultaneously, there was a strong synergistic effect between FcεRI and IL-33 (Figure 4a). This increase by combined (over individually applied) stimuli remained detectable in the presence of all inhibitors yet to very different extents. NF-κBi had the least influence, followed by JNKi, PI3Ki, and p38i, the latter three exerting similar effects (Figure 4c–f). Again, ERKi was most influential when TNF was elicited with the combination of IgER-CL plus IL-33. The order of importance of the distinct modules was, therefore, ERK >> p38 > PI3K ≈ JNK > NF-κB (brown columns in Figure 4). The order can also be seen in Appendix A, where ERKi and p38i reached significance, while other inhibitors showed tendencies in this multi-group comparison.

As demonstrated above, NF-κB and p38 are chiefly stimulated by IL-33, while activation of ERK and PI3K results primarily from FcεRI aggregation (Figure 2 and Figure 3). The results for p38i and NF-κBi (and, to a certain degree, JNKi) therefore suggest that the modules activated by IL-33 (p38, JNK, NF-κB) are not only independent from each other (as reported [50]), but that they can still contribute to synergy with other receptors when one of the three is blocked. This is similar for PI3K (only activated by IgER-CL), whose inhibition in the combined setting dampens production to the same level as p38i (mainly activated by IL-33) (Appendix A). The outcome is different for ERK, whose function is not only essential to TNF triggering by FcεRI singly (Figure 1b versus a) but also for its further potentiation by IL-33 (Figure 4b versus 4a and Appendix A). This emphasizes the importance and non-redundance of the ERK pathway for cytokine stimulation in skin MCs, as observed in previous work [68,76].

### 3.5. Synergism between IL-33 and MRGPRX2

TNF levels were also substantially elevated on combined stimulation between c48/80 and IL-33 over singly provided stimuli (Figure 5a), confirming our previous findings [50]. Here, the focus was on the mechanistic underpinnings underlying this synergy.

An increase by the combination over IL-33 was not detectable in the presence of ERKi or PI3Ki, suggesting that under these circumstances, expression was mainly driven by IL-33 without substantial contribution from c48/80 (Figure 5b,f). This is plausible considering that c48/80 elicits ERK1/2 and PI3K activation (Figure 3) and that both kinases, especially ERK, are involved in c48/80-mediated cytokine production [76]. In the presence of JNKi, synergy was still observable, as TNF levels were increased for the combination vis-à-vis single stimuli (Figure 1c). This was similar for p38i (Figure 5d). Thus, while stimulation in response to IL-33 alone was almost abolished by p38i (Figure 1d), potentiation by IL-33 (over c48/80 alone) was still noted under combined stimulation (Figure 5d). However, with p38 inhibition, the relative contribution of c48/80 increased, and the ratio of “combination:single stimulus” was higher against IL-33 than against c48/80. The same applied to NF-κBi, for which prominent potentiation of the stimuli was observed, even though NF-κBi countered TNF stimulation by IL-33 alone. The results for p38i and NF-κBi further emphasize that the single modules activated by IL-33 (p38, NF-κB) can contribute to the synergy, even if one is suppressed. Apparently, when combined with another potent stimulus, e.g., c48/80, any IL-33 module will suffice to elevate the response over that other stimulus. It will, therefore, be interesting to assess combinations of inhibitors in the future, including p38i + NF-κBi. Contrast to the pathways stimulated by IL-33 in the first place, the situation is different for the modules chiefly activated by c48/80 (i.e., ERK and PI3K), which are obviously both required for the potentiation over IL-33 to be statistically detectable. This is also confirmed in the depiction in Appendix A, whereby ERKi, PI3Ki, and p38i reached significance when the stimulus was IL-33 + c48/80. Overall, the order of significance of the distinct modules for cells co-activated by the two stimuli was ERK > PI3K ≈ p38 > JNK > NF-κB (grey columns in Figure 5 and Appendix A).

We conducted the same analysis replacing c48/80 with SP. Although SP alone did not elevate TNF expression over baseline (Figure 1), it did synergize with IL-33, as the combination of IL-33 + SP yielded greater production compared not only to SP, but also to IL-33. Synergy was not maintained in the presence of inhibitors with the exception of NF-κBi. Thus, in line with the findings for c48/80, there was no increase of IL-33 + SP over IL-33 under ERK or PI3K inhibition. Under these circumstances, TNF stimulation is driven by IL-33 alone, while SP cannot contribute if the two major modules it activates (ERK and PI3K) are suppressed. The presence of JNKi, and, more strongly, p38i, likewise abolished increase over IL-33. This was different for SP compared to c48/80. The result is, however, plausible considering that the increase over IL-33 in the combined situation was substantially weaker for IL-33 + SP (Figure 6) compared to IL-33 + c48/80 (Figure 5). Thus, the greatest fraction of TNF expression resulted from IL-33 stimulation when combined with SP, while the stimuli were equipotent on combination with c48/80. The order of modules relative to their significance for cells co-activated by SP and IL-33 was p38 >> ERK > JNK > NF-κB > PI3K, thus emphasizing the dominance of the IL-33 activated pathways over those elicited by SP (green columns in Figure 6). This is confirmed in the other type of data display, where inhibition was reached for p38i and pERKi in cells stimulated by this combination (Appendix A).

In aggregate, the findings for c48/80 (and less for SP) emphasize the significance of the PI3K and, especially, the ERK module when combined with IL-33, while the IL-33-activated modules (p38, NF-κB) are typically less prominent; therefore, blocking them singly will not abolish synergism (with the exception of SP + IL-33 wherein most of the TNF results from IL-33 stimulation). The findings are also similar for FcεRI + IL-33 (Figure 4) except for the greater dominance of ERK over PI3K (see also Appendix A).

### 3.6. NF-κB and Kinase Activation under Combinatorial Stimulation Shows no Influence of IL-33-Driven Modules by MRGPRX2 or FcεRI, but Slight Extension of pERK or pAKT by IL-33

As shown in Figure 2 and Figure 3, IL-33 initiated NF-κB and p38 signaling, while MRGPRX2 and FcεRI barely activated NF-κB in skin MCs but elicited pERK and pAKT instead. It was, therefore, important to realize whether the two receptor networks operate independently or cross-regulate the strength or duration of events elicited by the other stimulus. We, therefore, finally sought to quantitate the distinct signaling modules on combined vis-à-vis individual stimulation.

Initially, we explored three time points, i.e., 1 min (maximum of MRGPRX2-elicited signals), 5 min (signals with all receptors still visible or starting to become visible), and 30 min. While prominent pERK and some pAKT were noted after 1 min following MRGPRX2-ligation, they were not influenced by co-stimulation with IL-33 (Appendix A, upper panel). Likewise, the signals from IL-33, while only weakly observable after 1 min (i.e., IκB, pp65, and pp38) were not strengthened by SP or c48/80 (Appendix A, upper panel). The picture was similar after 5 min despite the much stronger signals of the IL-33-activated modules. FcεRI aggregation was also included in the 5 min point, not affecting the events elicited by IL-33 (IκB, pp65, pp38). Likewise, pERK and pAKT induced via MRGPRX2 or IgER were barely affected by IL-33 at this point; there was only a hint for pAKT on combined treatment with IL-33 and IgER-CL (Appendix A, lower panel).

In preliminary experiments, the strongest effects were found at 30 min, and we decided to add sufficient repetitions at this point to allow for rigorous statistical analyses to answer the question of whether signals can be prolonged by the other receptor network.

As shown in Figure 7, this was still not the case for IL-33 elicited modules. With all three proteins, i.e., IκB, pp65, and pp38, signals resulted from IL-33 stimulation alone and were not significantly modulated by either FcεRI-aggregation, c48/80, or SP (Figure 7a). Representative blots visualizing this aspect can be found in Figure 7b (upper three panels). The situation was somewhat different for the modules activated by FcεRI and MRGPRX2, namely, pERK and pAKT. Here, we found an enhancing effect of IL-33 on pERK elicited by MRGPRX2 with both ligands (Figure 7a), while no sustaining effect was noted after FcεRI-aggregation, perhaps due to the much higher signal observable at this point following FcεRI triggering alone (Figure 7b, fourth panel). Note, however, that no saturation of the signal was reached in these blots, excluding that the negative result (i.e., no reinforcement of pERK by IL-33) was due to technical issues. Conversely, pAKT was strengthened by IL-33 after stimulation via FcεRI, but not via MRGPRX2, the latter reflecting the weakness of signals (close to noise) detectable at 30 min in the pseudo-allergic setting (Figure 7b, fifth panel).

We conclude that there is no or modest cross-regulation across IL-33/ST2, FcεRI, and MRGPRX2-elicited signaling events. Moderate IL-33-mediated signal extension can be detected for pERK and pAKT. We deduce that the potent co-operation between IL-33 and FcεRI or MRGPRX2 results from the complementary nature of the activated modules in the first place, whereby ERK (mainly activated by FcεRI/MRGPRX2) and p38 (primarily activated by IL-33) assume the most significant roles.

## 4. Discussion

Despite numerous reports on cytokine stimulation in MCs, the molecular details that contribute to their production across different stimuli remain incompletely understood, also as the result of intricate networks encompassing feed-forward and feedback loops. The issue is even less clear in human MCs, since murine systems, especially bone marrow-derived mast cells (BMMCs), are the most commonly utilized in vitro MC model. However, BMMCs have been shown to deviate substantially from tissue MCs, even on comparison with other mouse MCs [77,78]. They also produce a distinct cytokine repertoire compared with human MCs. The clarification of signal transduction cascades initiated by diverse receptor networks likewise relied on BMMCs or MC lines in the past. Events elicited by receptor ligation in physiologic MCs from human tissues are still poorly defined though recent efforts are steadily reversing this lack of information [20,32,50,67,68,76,79,80,81,82,83]. These efforts have made it clear that cascades can differ between human and murine MCs (e.g., [84]).

In global terms, and this is largely independent of the species, signaling modules involved in the production of cytokines encompass the MAPKs ERK, p38, JNK, the PI3K/AKT pathway, PKC, and the transcription factors AP1, Calcineurin/NFAT, and NF-κB, each to a variable degree and in different constellations.

In particular, NF-κB is commonly associated with pro-inflammatory cytokines like TNF, IL-6, and IL-8. While NF-κB constitutes a major pathway activated by IL-33 also in human skin MCs, it remained unexplored whether it is activated and contributes to inflammatory genes on stimulation via FcεRI or MRGPRX2. We now show that, if at all, this occurs very inefficiently. In fact, neither IκB degradation, the seminal event in NF-κB activation, nor p65 phosphorylation nor suppression of TNF production by NF-κBi were observed for MCs activated by the FcεRI- or the MRGPRX2-route. Conversely, NF-κB was potently activated by IL-33 in the same cells inspected concurrently. The minor or absent relevance of NF-κB in skin MCs has important corollaries. It shows that NF-κB is not invariably activated across stimulatory receptors, especially those eliciting clinically relevant allergic and pseudo-allergic responses. Therefore, inhibitors directed at the NF-κB module will be unlikely successful in diseases initiated or perpetuated by MC cytokines triggered by FcεRI or MRGPRX2. In these settings, the targeting of ERK will be far more effective.

Activation of NF-κB following FcεRI aggregation occurs in other types of MCs, though, especially in transformed or immature subsets like RBL-2H3 cells and BMMCs, where it underlies cytokine expression, including of TNF [85,86,87,88]. This further underlines the discrepancy across MC subsets. One of the adaptors to regulate NF-κB, p38, and JNK alike following FcεRI aggregation in murine MCs is TRAF6 [88]. In human skin MCs, not only NF-κB activation does not occur, but p38 and JNK are likewise weakly stimulated upon FcεRI and MRGPRX2 ligation. The inefficiency in skin MCs may, therefore, stem from the inability to stimulate TRAF6 in the first place. In addition, or alternatively, NF-κB activation downstream of FcεRI has been described to depend on Bcl10 and Malt1 in the mouse [89]. Expression of these entities is low in skin MCs across the comprehensive FANTOM5 atlas, while much higher levels are detectable in other myeloid cells [10,11,12]. Dependency on weakly expressed Bcl10 and Malt1 may potentially explain the discrepancy between human and mouse MCs. Further studies will be required to clarify this issue. Of note, while the PI3K/AKT network triggered by FcεRI aggregation can, in principle, promote TNF production through an NF-κB-independent pathway, it is also a potent elicitor of NF-κB [90]. Again, while AKT activates NF-κB in murine MCs [90], this connection seems unproductive in human skin MCs, since the relatively robust activation of AKT downstream of FcεRI does not translate to NF-κB activation in these cells.

NF-κB activation by FcεRI is less established for human MCs. In CD34+ in vitro generated MCs, at least, p65-phosphorylation was detectable [91]. In intestinal MCs NF-κB was involved in the production of TNF and several other cytokines following FcεRI-aggregation, but the conclusion was based on one inhibitor (curcumin, which also has other effects) and not on immunoblot results [92]. Importantly, even in intestinal MCs, the dominant effect elicited by FcεRI was FOS induction, which is, likewise, a key event downstream of several activation axes in skin MCs and requires ERK-mediated activation of CREB [68,82,83]. In human lung MCs, IκB degradation and NF-κB nuclear recruitment were evaluated by microscopy upon anti-IgE + SCF stimulation, whereby some delayed degradation was detected, and TNF was identified as an intermediary in this setting [93]. This speaks in favor of a weak or absent NF-κB pathway in FcεRI-activated human lung MCs; instead, it requires TNF released from MCs rather than resulting from FcεRI-initiated signaling itself. In conjunction with our findings, all aggregated data imply that NF-κB activation by FcεRI is a minor component in human tissue MCs compared to murine MCs (and compared to human MCs stimulated by IL-33).

Due to its more recent discovery, signaling pathways downstream of MRGPRX2 are less understood compared to FcεRI. We reported on potent ERK activation and ERK requirement for cytokine generation in skin MCs [76]. Regarding NF-κB, LL-37 (another ligand of MRGPRX2 and its mouse ortholog Mrgprb2) induced this module in mouse peritoneal MCs, obviously requiring beta-arrestin-2 [94]. In the same cells, TNF and IL-6 were both induced by SP through several signaling pathways, including NF-κB, whose activation was demonstrated by translocation, and inhibitors [95].

In the human LAD2 cell line, NF-κB was induced by SP [96]. Our data further underline the discrepancy between LAD2 and skin MCs regarding signaling and functional outputs downstream of MRGPRX2; in fact, LAD2 cells show hyperactivity towards MRGPRX2 compared with physiologic MC subsets (summarized in [34]). Our group currently investigates the underpinnings behind the profound differences between these cell types.

Overall, we conclude that in human skin MCs, potent NF-κB activation requires receptors other than FcεRI or MRGPRX2, and that activation of NF-κB by FcεRI or MRGPRX2 is limited to murine MCs and cell lines.

Apart from dissecting the role of NF-κB in skin MCs, the current study could confirm the crucial character of ERK for cytokine generation stimulated via FcεRI or MRGPRX2; in fact, ERK was the most central module, followed by PI3K and JNK [76]. Conversely, p38 and NF-κB were the most important modules in IL-33-triggered cytokine production (Figure 1). And p38′s crucial role in TNF generation elicited by IL-33 is in accordance with previous data [50]. It remains to be determined whether the p38-activated kinases MK2 and MK3 (MAPK-activated protein kinases 2 and 3) are relevant effectors of cytokine induction in human MCs as they are in the mouse [9]. Collectively, the principal modules differ between IL-33 on the one hand and FcεRI/MRGPRX2 on the other hand.

Again, differences also exist between human and rodent MCs regarding MRGPRX2/b2 signaling. While in human skin MCs, signaling was rapidly activated with varied efficiency across kinases in the order of pERK1/2 > pAKT > pp38 > pJNK [76], Azzolina et al. found swift phosphorylation of p38 and JNK but substantially delayed ERK1/2 phosphorylation in rat peritoneal MCs following SP stimulation [97]. In addition, while ERK did not play a role in their system, p38 was involved in SP-triggered TNF in rat MCs, while in our study, p38 was dispensable (Figure 1). These discrepant findings substantiate that findings in rodents cannot be easily extrapolated to humans and should be interpreted with caution.

In addition to exploring the significance of distinct components in MCs activated individually by the allergic, the MRGPRX2-dependent, or the IL-33-mediated route, a major focus of the study was to assess how signaling modules join forces when different receptors are activated in sync. The significance of distinct modules in combinatorial scenarios is more complex and even less comprehended. There are two major (not mutually exclusive) possibilities of how combinatorial stimulation may be organized. 1. Strengthened or prolonged activation of the same modules. In this scenario, the activated modules do not need to differ between the two receptors in question. 2. Each receptor stimulates its individual components without much interaction with those activated by the other system. Synergism is then brought about by the mixture/complementation of modules. Since signaling pathways are sufficiently distinct between IL-33 versus FcεRI/MRGPRX2, option 2 appeared feasible, without discarding option 1 entirely.

To explore this comprehensively, we used Western blot analyses, and multiple selective inhibitors. Since we work with primary MCs from human skin, it was also important to run experiments sufficient times to find patterns and draw meaningful conclusions. It became obvious that option 2 is likely dominant. In fact, by Western blot analysis, the IL-33-elicited components (p38, NF-κB) remained unaffected by additional stimulation of FcεRI/MRGPRX2 (Figure 7 and Appendix A). This was somewhat different for FcεRI/MRGPRX2-elicited signals (pERK, pAKT), which were prolonged by IL-33, though also quite moderately (Figure 7 and Appendix A). Notwithstanding, the augmentation of pERK upon IL-33 plus MRGPRX2 agonists observed here was similarly reported for IL-33 plus anaphylatoxins (over anaphylatoxins alone) and may, therefore, apply to different GPCRs [55]. In contrast, no such increase was observed for FcεRI-elicited ERK phosphorylation. Conversely, pAKT was exclusively strengthened by IL-33 upon activation of FcεRI. This raises the interesting possibility that, despite similar components being activated by FcεRI and MRGPRX2, IL-33 can identify what receptor this activation originated from to differentially stabilize these events. However, it also demonstrates that the prolongation of the ERK signal is unlikely of major importance, since the augmentation of TNF by IL-33 (and dominance of ERK in this augmentation) was similar for FcεRI as for MRGPRX2 (triggered with c48/80). Nonetheless, the extended ERK activation may be contributory, at least to some degree, when ST2 and MRGPRX2 are simultaneously triggered.

Under combinatorial stimulation, synergism at TNF production was also still observable in the presence of inhibitors, which would not have been expected if one essential module (such as p38 for IL-33) had been strongly increased and/or prolonged by the other receptors (FcεRI/MRGPRX2). The closest to elimination of the TNF response was ERKi, resulting from the nearly complete abrogation of FcεRI/MRGPRX2-mediated stimulation and some reduction also of the IL-33-mediated route (not reaching significance in the IL-33-only setting though; Appendix A). However, even under ERKi the response to “IL-33 plus other stimulus” was increased over “other stimulus” in all cases (FcεRI aggregation, SP, c48/80). Therefore, even in this extreme situation, IL-33 could still elevate the nearly eradicated response to “other stimulus”. Co-operation was also detectable in basically all other inhibitor/stimulus combinations, whereby significance was achieved at least against one stimulus, typically against the one that elicited the suppressed module in the first place. To give an example, in the presence of PI3Ki, there was an enhancement of SP + IL-33 and c48/80 + IL-33 over SP or c48/80 alone, but not against IL-33, since the contributions of SP and c48/80 were inhibited, while IL-33′s action was unperturbed. In addition, a modular contribution of the distinct pathways downstream of one receptor could be detected. For example, while p38 and NF-κB were both relevant for IL-33 to convey signals to the TNF-producing machinery, blocking either did not completely abrogate the co-operation between IL-33 and “other stimulus”, and IL-33 still elevated the response to “other stimulus” in nearly all cases. Thus, the more cascades of similar potency are activated, the lesser the need for any single one of them, as exemplified by p38/NF-κB when combined with ERK (and/or PI3K).

Collectively, the strong co-operation between IL-33 and FcεRI/MRGPRX2 can best be explained by the complementary nature of the signaling pathways the receptor systems activate, i.e., strong ERK, PI3K, and little JNK in the case of MRGPRX2 and FcεRI (but little p38, and no NF-κB), but profound p38, some JNK, and substantial NF-κB (but little ERK, and no PI3K) in the case of IL-33. The distinct modules, thus, seem to operate in a largely independent fashion and combine only in the final steps.

These findings are of pathophysiological relevance since combinatorial stimulation is more common in the cells’ natural habitat where MCs are exposed to different cues simultaneously. For instance, cPLA2 contained in hymenoptera venoms, i.e., potent allergens (which also comprise MRGPRX2 ligands like mellitin or mastoparan), can additionally liberate IL-33 from endothelial cells [98]. The same applies to house dust mites, which are not only allergens activating FcεRI but by virtue of their protease activity, can liberate neuropeptides that will activate MRGPRX2/b2 to induce or increase MC activation [99]. Importantly, IL-33 has been associated with skin diseases like psoriasis and contact dermatitis [100,101,102,103]. There is an especially close link between IL-33 and atopic dermatitis: while efficacy of anti-IL-33-tageted therapies is still not completely defined, etokimab (a therapeutic anti-IL-33 antibody) showed improvement in skin score and reduced skin neutrophil infiltration after a single administration [104,105]. Systemic conditions like food allergy and anaphylaxis also seem to be supported by IL-33 [106,107]. On the other hand, despite relatively recent discovery, evidence has accumulated to suggest important contributions of MRGPRX2 to skin diseases like AD, psoriasis, and anaphylaxis, the latter especially when induced by drugs [37,108,109,110,111]. Therefore, since combined stimulation of MCs by IL-33 and the pseudo-allergic/neurogenic route is likely frequent, their crosstalk will require further understanding both experimentally and clinically. The current study provides insights into signal integration, laying some fundamental groundwork also for therapeutic considerations.

## 5. Conclusions

MRGPRX2, FcεRI, and IL-33/ST2 on MCs likely act in co-operation when these are embedded in their native cutaneous environment. IL-33 strongly synergizes with MRGPRX2, or FcεRI, enhancing their potential to elicit biologically meaningful reactions. This study shows that in human skin MCs, IL-33 operates chiefly via p38 and NF-κB (followed by JNK), while the partner receptor operates via ERK and PI3K in the first place. The complementary nature of activated modules between IL-33 on the one hand and MRGPRX2/FcεRI on the other is the key mechanism behind synergism. IL-33, thus, enables MRGPRX2/FcεRI function by providing additional signaling modules, establishing a cytokine-favoring environment in the skin habitat that contributes to inflammation and disease. While inhibition of individual modules downstream of IL-33 reduces cytokine expression, it does not completely abrogate the process under joint activation with MRGPRX2/FcεRI. Conversely, ERK is verified as the most significant contributor to skin MC cytokines downstream of several receptors, while ERK’s function seems to be taken over by p38 in the context of the IL-33/ST2 axis. These findings have relevance to clinical settings in which MC-derived cytokines are significant drivers of the pathology.

## Figures and Tables

**Figure 1 cells-12-02700-f001:**
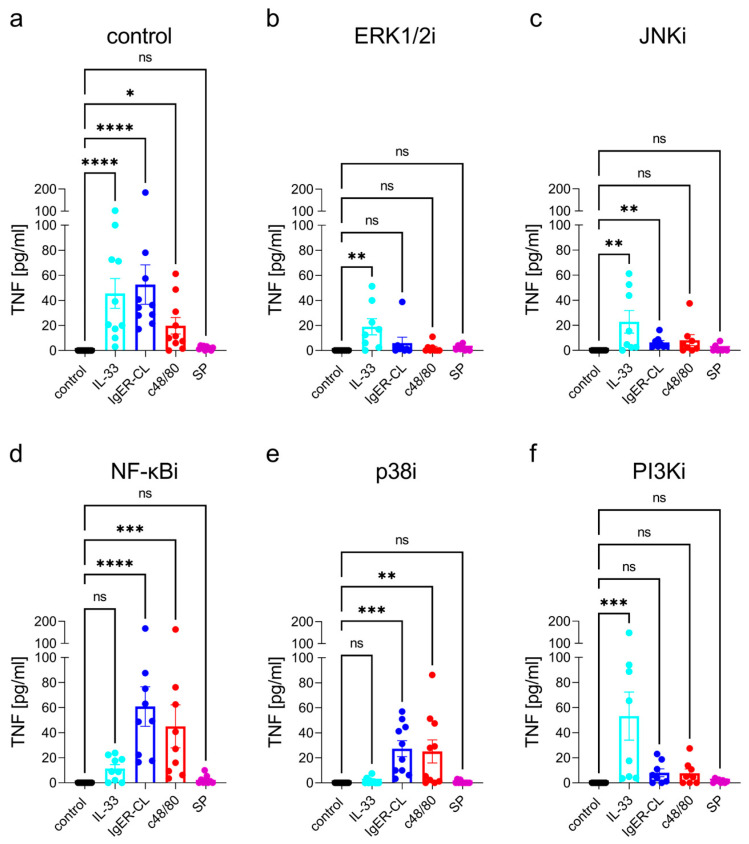
Distinct modules contribute to TNF stimulation in dependence of the stimulus. Skin MCs were stimulated by IgER-CL (crosslinking, AER-37 at 0.2 µg/mL), or SP (30 µM), or c48/80 (10 µg/mL), or IL-33 (20 ng/mL) for 24 h, TNF was quantified in the resulting supernatants by ELISA. (**a**) control; vehicle was used instead of inhibitor. b-f, cells were pretreated with the respective inhibitors for 15 min and then stimulated as in (**a**). (**b**) ERK1/2 inhibitor SCH772984 at 10 µM, (**c**) JNK inhibitor SP600125 at 10 µM, and (**d**) p38 inhibitor SB203580 at 10 µM. (**e**) NF-κB inhibitor BAY11-7082 at 1 µM and (**f**) PI3K inhibitor Pictisilib at 10 µM. Mean ± SEM of *n* = 8–10 separate experiments; individual values are given as dots. The *y*-axis is consistent across inhibitors to allow direct comparison. * *p* < 0.05, ** *p* < 0.01, *** *p* < 0.001, **** *p* < 0.0001, ns: not significant. Friedman test with Dunn’s multiple comparisons test. i = inhibitor.

**Figure 2 cells-12-02700-f002:**
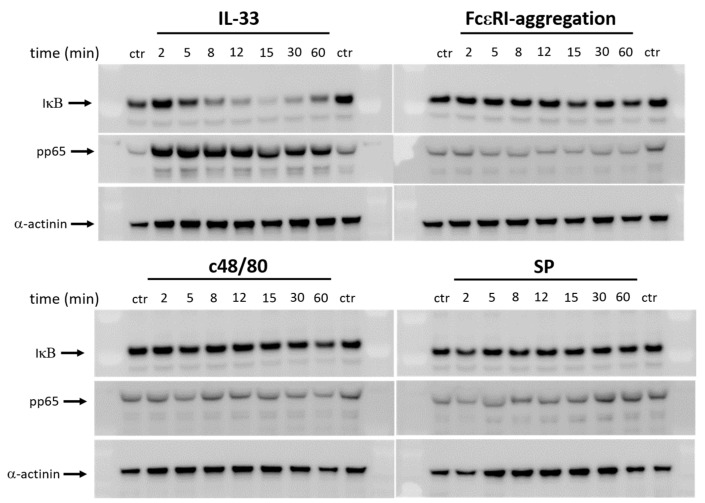
IL-33 elicits the NF-κB pathway in human skin MCs, while FcεRI and MRGPRX2 do not. MCs were stimulated by IL-33 (20 ng/mL), IgER-CL (crosslinking, AER-37 at 0.2 µg/mL), or SP (30 µM) or c48/80 (10 µg/mL) for the times given and degradation of IκBα and phosphorylation of p65 were detected by immunoblot. α-actinin served as the loading control. And 1 out of 3 experiments with comparable outcome is shown.

**Figure 3 cells-12-02700-f003:**
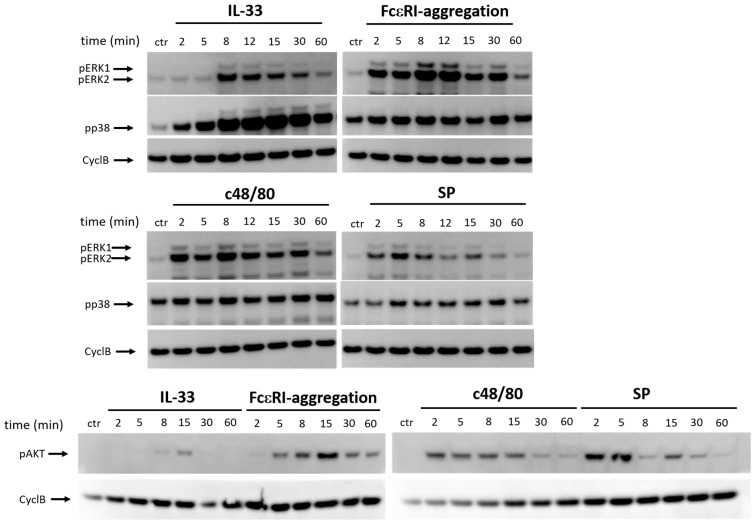
IL-33 potently activates p38, while FcεRI and MRGPRX2 elicit pERK and pAKT. Skin MCs were stimulated by IL-33 (20 ng/mL), IgER-CL (crosslinking, AER-37 at 0.2 µg/mL), or SP (30 µM), or c48/80 (10 µg/mL) for the times given and phosphorylations of ERK1/2, p38 and AKT were detected by immunoblot. Cyclophilin B (CyclB) served as the loading control. Note that pERK activation is only occasionally visible after IL-33 stimulation. And 1 out of 3 experiments with largely comparable outcome is shown.

**Figure 4 cells-12-02700-f004:**
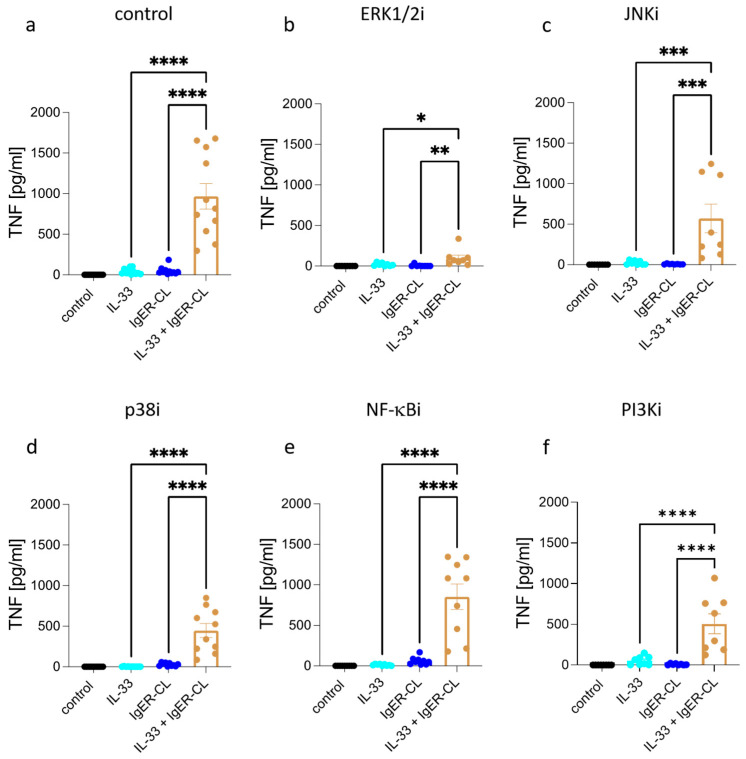
Synergism between IL-33 and FcεRI aggregation relies on IL-33 and FcεRI-stimulated modules with a dominance of ERK1/2. Skin MCs were stimulated by IgER-CL (crosslinking, AER-37 at 0.2 µg/mL), IL-33 (20 ng/mL), or both combined for 24 h; TNF was quantified in the resulting supernatants by ELISA. (**a**) control; vehicle was used instead of inhibitor. b-f, cells were pretreated with the respective inhibitors for 15 min and then stimulated as in (**a**). (**b**) ERK1/2 inhibitor SCH772984 at 10 µM, (**c**) JNK inhibitor SP600125 at 10 µM, and (**d**) p38 inhibitor SB203580 at 10 µM. (**e**) NF-κB inhibitor BAY11-7082 at 1 µM and (**f**) PI3K inhibitor Pictisilib at 10 µM. Mean ± SEM of *n* = 10 separate experiments; individual values are given as dots. The *y*-axis is consistent across inhibitors to allow direct comparison. * *p* < 0.05, ** *p* < 0.01, *** *p* < 0.001, **** *p* < 0.0001. Friedman test with Dunn’s multiple comparisons test. i = inhibitor.

**Figure 5 cells-12-02700-f005:**
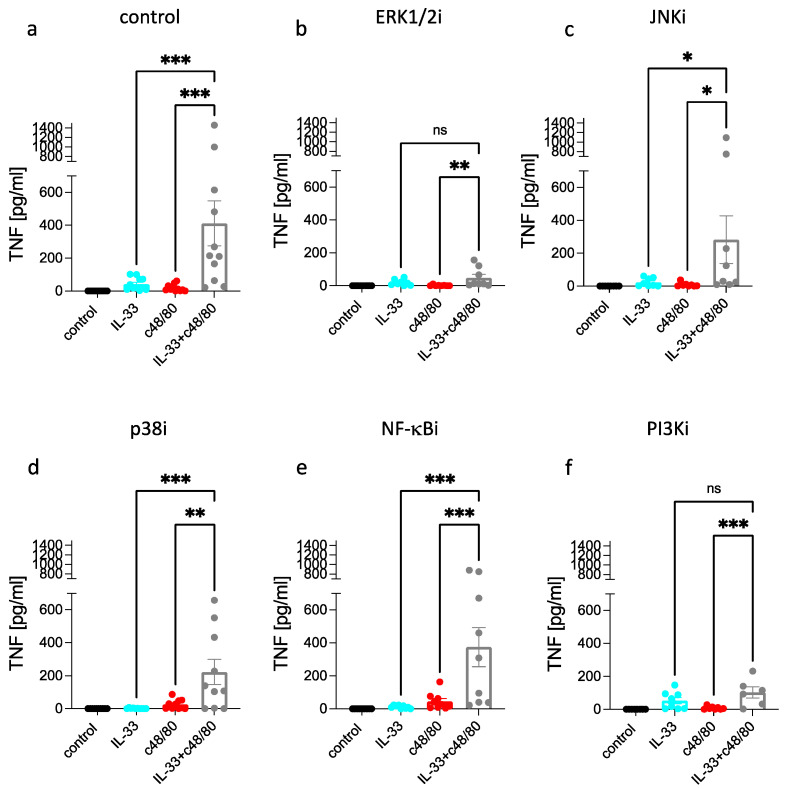
The synergism between IL-33 and c48/80 most strongly relies on ERK activity followed by other modules. Skin MCs were stimulated by c48/80 (10 µg/mL), IL-33 (20 ng/mL), or both combined for 24 h; TNF was quantified in the resulting supernatants by ELISA. (**a**) control; vehicle was used instead of inhibitor. (**b**–**f**), cells were pretreated with the respective inhibitors for 15 min and then stimulated as in (**a**). (**b**) ERK1/2 inhibitor SCH772984 at 10 µM, (**c**) JNK inhibitor SP600125 at 10 µM, and (**d**) p38 inhibitor SB203580 at 10 µM. (**e**) NF-κB inhibitor BAY11-7082 at 1 µM and (**f**) PI3K inhibitor Pictisilib at 10 µM. Mean ± SEM of *n* = 10 separate experiments; individual values are given as dots. The *y*-axis is consistent across inhibitors to allow direct comparison. * *p* < 0.05, ** *p* < 0.01, *** *p* < 0.001, ns: not significant. i = inhibitor.

**Figure 6 cells-12-02700-f006:**
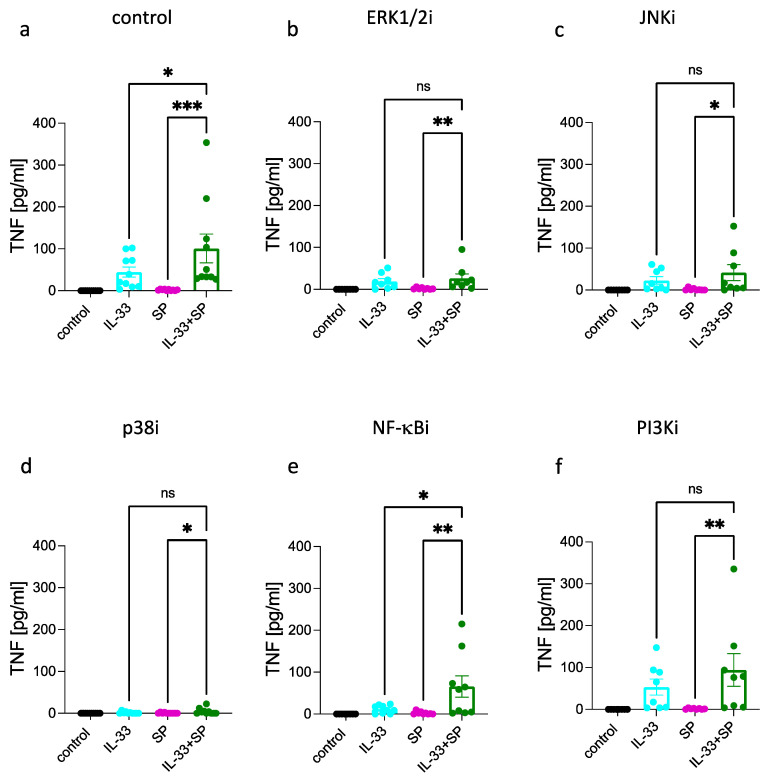
The modest synergism between SP and IL-33 relies most strongly on p38, followed by ERK. Skin MCs were stimulated by SP (30 µM), IL-33 (20 ng/mL), or both combined for 24 h; TNF was quantified in the resulting supernatants by ELISA. (**a**) control, vehicle was used instead of inhibitor. b–f, cells were pretreated with the respective inhibitors for 15 min and then stimulated as in (**a**). (**b**) ERK1/2 inhibitor SCH772984 at 10 µM, (**c**) JNK inhibitor SP600125 at 10 µM, and (**d**) p38 inhibitor SB203580 at 10 µM. (**e**) NF-κB inhibitor BAY11-7082 at 1 µM and (**f**) PI3K inhibitor Pictisilib at 10 µM. Mean ± SEM of *n* = 10 separate experiments; individual values are given as dots. The *y*-axis is consistent across inhibitors to allow direct comparison. * *p* < 0.05, ** *p* < 0.01, *** *p* < 0.001, ns: not significant. Friedman test with Dunn’s multiple comparisons test. i = inhibitor.

**Figure 7 cells-12-02700-f007:**
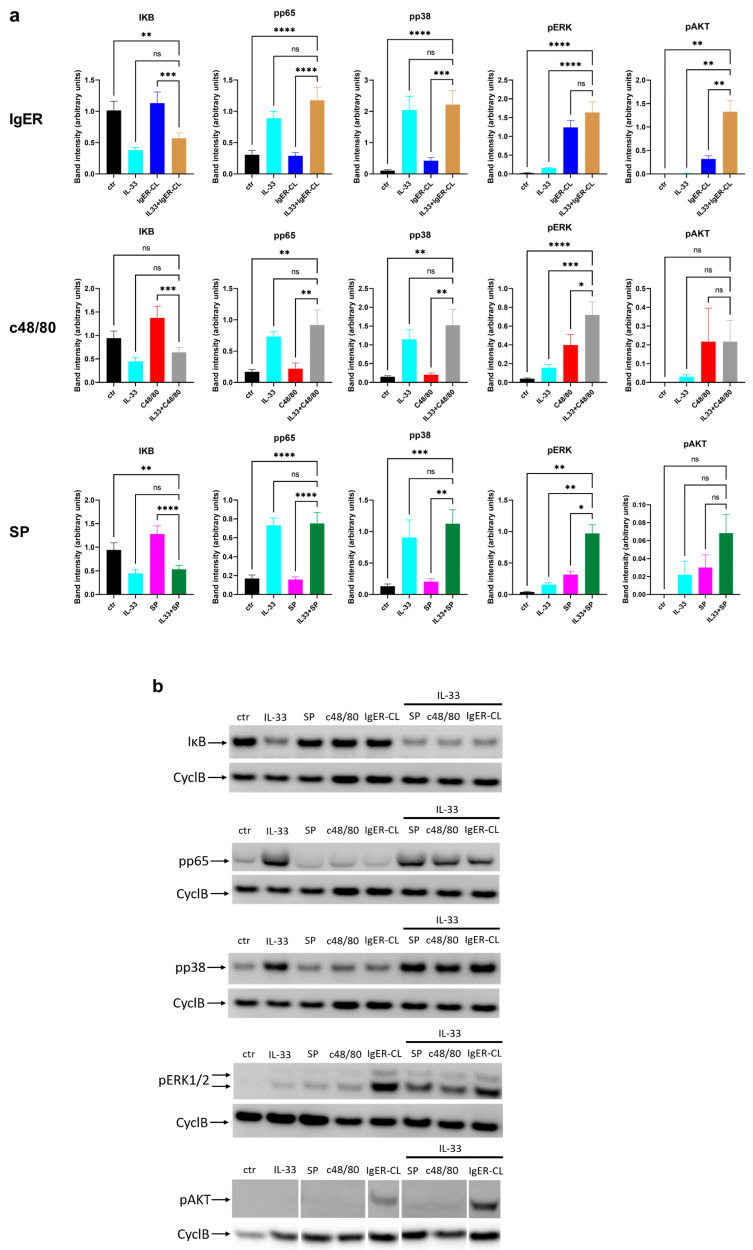
IL-33, FcεRI, and MRGPRX2 triggered signaling events only modestly influence each other. Skin MCs were stimulated by IL-33 (20 ng/mL), IgER-CL (crosslinking, AER-37 at 0.2 µg/mL), SP (30 µM), or c48/80 (10 µg/mL), or the specified combinations for 30 min. The degradation of phosphorylation of signaling components were detected by immunoblot. Cyclophilin B (CyclB) served as the loading control. (**a**) Mean ± SEM of *n* = 6–11. (**b**) Representative individual blots. * *p* < 0.05, ** *p* < 0.01, *** *p* < 0.001, **** *p* < 0.0001, ns: not significant. RM one-way ANOVA with Holm-Šídák’s multiple comparisons test.

## Data Availability

Data are contained within the article and Appendix A.

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
