# Peer review of "Synergism between IL-33 and MRGPRX2/FcεRI Is Primarily Due to the Complementation of Signaling Modules, and Only Modestly Supplemented by Prolonged Activation of Selected Kinases"

_cells, 2023, doi:10.3390/cells12232700_

Round 1
Reviewer 1 Report
Comments and Suggestions for Authors
Please find my comments in the attached pdf file.

Reviewer 2 Report
Comments and Suggestions for Authors
This is a very interesting paper which describes the effects of combinatory stimulations of human MCs. However, there are some technical critics.
The inhibitors should be tested for their cytotoxicity
In all ELISA experiments the stimulations without inhibitors are always shown separately from the stimulations with inhibitors. Therefore, there is no statistical comparisons between the effects without inhibitors and the effects with inhibitors shown. How can the authors conclude that the inhibitors reduce or block the TNF production without direct statistical comparisons?
Fig 3 loading control (CyclB) follow the pAkt phosphorylation. Therefore, it is not clear whether there is a real activation of pAKT after IL-33 stimulation.
In Fig7a in some diagrams the statistics are missing.
In Fig7B the pAKT blot should be repeated in order to show one blot for all conditions.
NFkB activation is induced by IL-33 which culminates in a NFkB mediated TNF production. However, in combination with IgE, NFkB inhibition did not reduce the resulting TNF production. How do the authors explain this? IgE also induces the activation of NFAT. Therefore, I would suggest to make ELISAS with cyclosporine A to determine the influence of NFAT.
Round 2
Reviewer 2 Report
Comments and Suggestions for Authors
The authors adequatly answered my questions and/ or improved the manuscript. This work is now ready for publication.
Author Response
I would strongly recommend this one experiment with both inhibitors together (i.e. combined presence of inhibitors of NFκBi and P38i), as I stated in my comments, as it is crucial for the main message of the paper.
We have taken care of this aspect and present the new set of data as a new Fig. S2, also pasted below. For the new experimental series, we selected the combinations IL-33+IgER-CL and IL-33+c48/80, since they elicited the highest levels of TNF, so that even small differences would become apparent. Combining p38i with NFkBi had an additional impact compared to the single inhibitors, especially in the combination with IgER-aggregation. Yet still, ERKi (included in this series for direct comparison) was more potent than p38i and NFkBi combined. We refer to these new data in Methods, Results and Discussion (highlighted in the MARKED version of the Ms and supplement).
